# Vaccination and Immunity toward Measles: A Serosurvey in Future Healthcare Workers

**DOI:** 10.3390/vaccines9040377

**Published:** 2021-04-13

**Authors:** Andrea Trevisan, Paola Mason, Annamaria Nicolli, Stefano Maso, Bruno Scarpa, Angelo Moretto, Maria Luisa Scapellato

**Affiliations:** 1Department of Cardiac Thoracic Vascular Sciences and Public Health, University of Padova, 35128 Padova, Italy; paola.mason.1@unipd.it (P.M.); annamaria.nicolli@unipd.it (A.N.); stefano.maso@unipd.it (S.M.); angelo.moretto@unipd.it (A.M.); marialuisa.scapellato@unipd.it (M.L.S.); 2Department of Statistical Sciences, University of Padova, 35128 Padova, Italy; bruno.scarpa@unipd.it; 3Department of Mathematics “Tullio Levi-Civita”, University of Padova, 35128 Padova, Italy

**Keywords:** measles, vaccination schedule, vaccination response, students, healthcare workers

## Abstract

Measles is a very contagious infectious disease, and vaccination is the only medical aid to counter the spread of the infection. The aim of this study was to evaluate the influence of vaccination schedule and type of vaccine, number of doses, and sex on the immune response. In a population of Italian medical students (8497 individuals born after 1980 with certificate of vaccination and quantitative measurement of antibodies against measles), the prevalence of positive antibodies to measles and antibody titer was measured. Vaccination schedule such as number of doses and vaccine type (measles alone or combined as measles, mumps and rubella (MMR)) and sex were the variables considered to influence the immune response. The vaccination schedule depends on the year of birth: students born before 1990 were prevalently vaccinated once and with measles vaccine alone (not as MMR). One dose of vaccine induces a significantly (*p* < 0.0001) higher positive response and antibody titer than two doses, in particular when measles alone is used (*p* < 0.0001). Females have a significantly higher percentage of positive response (*p* = 0.0001) than males but only when the MMR formulation was used. Multiple linear regression confirms that sex significantly influences antibody titer when only MMR is used, after one (*p* = 0.0002) or two (*p* = 0.0060) doses. In conclusion, vaccination schedule and, partially, sex influence immune response to measles vaccination. Most notably, the measles vaccine alone (one dose) is more effective than one and two doses of MMR.

## 1. Introduction

The vaccination toward measles (one dose schedule) was introduced in Italy in 1976 and implemented since 1979 [1]; the combined measles, mumps, and rubella (MMR, rather two doses) vaccine was introduced in 1999 [2] and approved by the National Plan for eradication of measles and congenital rubella [3], according to the objectives of World Health Organization 2012–2020 [4]. Since 7 June 2017 MMR is mandatory in Italy [5] for newborns and adolescents until 16 years of age. In addition, the Italian National Vaccination Prevention Plan (NVPP) 2017–2019 [6] strongly suggests that healthcare workers (HCWs) be vaccinated toward seven transmittable diseases among which MMR. On the same track, the Italian Multidisciplinary Society for Infection Prevention in Health Organizations, the Italian Society of Occupational Health, the Italian Society of Hygiene, and others have drawn up the so-called “Pisa card” [7] signed during the Pisa conference on 27–28 March 2017 related to the vaccination of HCWs (currently excluded by mandatory vaccination).

Vaccinations for HCWs are an important issue, and in the face of hesitancy phenomena, there is discussion on the need for at least those indicated by the NVPP and the Pisa card to be made mandatory. In these days, in the middle of the COVID-19 pandemic, there is discussion about the need for the vaccine to be mandatory for HCWs.

The current vaccination schedule for MMR provides a first dose in the second year of life and, according to elimination plane [3], a second dose at 5–6 (recommended) or 11–12 years of age. The vaccination coverage for MMR reached in Italy a peak in 2010 (90.6%), falling short to the critical rate for measles to reach the herd immunity greater than 95% [8].

It has been recognized that the effectiveness of the first dose of measles vaccine is more than 95%, increasing to more than 99% after the second dose; the immunity persists for a long time [9]. A recent study reports that 15 years after the second dose of MMR vaccine, the rate of measles seropositivity was around 95%. According to this, the two-dose schedule is strongly recommended [10].

Our previous research [11,12,13,14] provided evidence that medical school students were not completely protected, at least as regards the rate of circulating antibodies.

The aim of this study was to thoroughly analyze the history of measles vaccination (age at vaccination, number of doses, vaccination schedule, time since the last dose) to evaluate vaccination coverage and immunization toward measles in a cohort of students belonging to the Medical School of Padua University (Padua, Italy).

## 2. Materials and Methods

### 2.1. Population

According to Italian law on safety and health at work [15], the students belonging to the degree courses of the Medical School of Padua University (medicine and surgery, dentistry, and healthcare professions) are submitted to health surveillance in the second (medicine and surgery until 2016 and dentistry) or the first (medicine and surgery since 2017 and healthcare professions) year of course. From 2004 until February 2020, 13,553 students had been screened measuring antibodies of transmissible, but preventable diseases. Enrollment criteria included (i) having been vaccinated against measles, (ii) having been born in Italy, and hence having likely the same vaccination schedules since 1980, as measles vaccination was implemented in Italy in 1979, (iii) availability of the certificate of vaccination released from the Public Health Office, and (iv) a quantitative measurement of antibodies against measles. According to these criteria (Figure 1), enrolled students were 8497 (62.7%) (2990 males and 5507 females, ratio males/females 0.54). Unvaccinated students (531, 5.9%) were excluded.

As it is specified in Table 1, the group of vaccinated subjects toward measles had a mean age of 21.1 ± 2.0 years without significant differences between males and females, and the geographical origins were prevalently from Northern Italy and Veneto Region.

### 2.2. Vaccination Schedule

Depending on the year of birth, the measles vaccination schedule was different. The vaccination certificates released by the Public Health Office demonstrate that measles vaccine was administered with one dose, measles alone or MMR, or with two doses (two doses of measles alone, a dose of measles alone plus one dose of MMR or two doses of MMR). 

### 2.3. Measurement of Measles Antibodies

The measles IgG antibodies were measured by means of commercial enzyme-linked immunosorbent assay (EIA) Enzygnost (Dade Behring, Marburg, Germany). Antibody levels of measles were reported as positive (higher than 350 IU/mL), negative (lower than 150 IU/mL), or equivocal (150–350 IU/mL). Equivocal results had been statistically processed as negative according to CDC recommendations [16].

### 2.4. Statistics

Chi-square (χ^2^) test 2 by 2 (Yates correction) was used to compare the prevalence of antibody positivity. Comparison between means was done with unpaired t-test, assuming unequal variances. We adjusted the results for multiple testing by considering the Bonferroni correction in which the *p*-values are multiplied by the number of comparisons. Multiple linear regression based on the logarithms of antibody titer (being asymmetric its distribution) was used to analyze the variables influencing antibodies titer (dependent variable), and the following outcomes were considered as independent variables: (1) sex, (2) age at first dose of vaccine, (3) time since the dose (or the second dose) of vaccine, and (4) typology of vaccine categorized as 1 (measles alone), 2 (MMR), 3 (measles alone plus measles alone), 4 (measles alone plus MMR) and 5 (two doses of MMR). Typology 1 is the reference type. Linear regression coefficient r (Pearson product-moment correlation coefficient) was used (if appropriate) to relate single independent variables with antibody titer. Furthermore, four years of birth groups were assumed as previously [14]: born in 1980–1985, 1986–1990, 1991–1995, and after 1995. Other statistical analyses are descriptive. Significance is stated by *p* < 0.05. Statsdirect 2.7.7 version (Statsdirect Ltd., Birkenhead, Merseyside, UK) has been used for statistical analyses.

## 3. Results

On the enrolled population attending medical school courses and born since 1980, 94.1% were vaccinated toward measles (22.4% one dose, 77.6% two doses), including significantly (*p* = 0.0041) more females (94.7%) than males (93.1%) (data not shown).

The vaccination schedule (Figure 2) is very different depending on the year of birth. One dose of vaccine was prevalently adopted before 1990, with a prevalence of measles alone before 1985. Children born after 1995 were covered by two doses of MMR vaccine.

By the vaccination certificate (data not shown), students vaccinated once received the vaccine dose at a significantly (*p* < 0.0001) later age as compared to those vaccinated twice (3.9 ± 4.2 and 1.9 ± 1.8 years of age, respectively), especially if vaccinated with the MMR vaccine. The timing and type of vaccination is dependent on year of birth. 

The most relevant result is that one dose induces a significantly (*p* < 0.0001) higher prevalence of positives antibodies and higher antibody titer than two doses, despite the longer (about double) time since the last dose; furthermore, the measles vaccine alone is most effective, and this is confirmed by the double use of measles alone that induces a similar high percentage of positives (96.7% vs. 96.6%) and antibody titer (Table 2).

Overall, females are more responsive than males to measles vaccine (Table 3) after one (*p* = 0.0004 positives and *p* = 0.0010 titer) or two doses (*p* < 0.0001 positives and *p* = 0.0110 titre), but only when MMR (once or twice) is administered. 

Multiple linear regression analysis highlights that all independent variables influence antibody titer, and it confirms that vaccine typology 1 (one dose of measles alone) is the most effective, followed by typology 3 and 4, while typology 2 (one dose of MMR) and 5 (two doses of MMR) appear to be the least effective (Table 4). Sex significantly (*p* < 0.0001) influences antibody titer but only when MMR was used. In addition, the more delayed the first dose, the better the antibody response, as also demonstrated by the linear correlation between the log of the antibody titer and the log of age at the first dose (*r* = 0.154, *p* < 0.0001). Finally, the time between vaccination and antibody titer analysis on multivariate analysis seems to have some influence, which is not confirmed by the linear regression between the two variables (*r* = 0.0058, *p* = 0.5924).

Finally, if stratified by year of birth groups, those born between 1980 and 1990 have a significantly (*p* < 0.0001) higher response in terms of positive antibodies and antibody titer than younger groups after one and two doses (Table 5). In addition, sex differences were statistically significant in a randomized manner between the groups.

The students never vaccinated against measles (531 individuals, data not shown) were excluded from the casuistry according to enrollment criteria, but they have a certain relevance. A significantly (*p* < 0.0001) high prevalence of positive antibodies (97.5% and 87.4%, respectively) and antibody titer (6263.6 ± 4340.9 and 6375.0 ± 4600.2 IU/mL, respectively) was observed in students born in 1980–1985 and 1986–1990 compared to those born in 1991–1995 (prevalence 32.5%, antibody titer 2291.1 ± 3882.1 IU/mL) and after 1995 (prevalence 14.3%, titer 386.7 ± 1012.6 IU/mL).

## 4. Discussion

Measles is probably the most contagious infectious disease for humans [17]. In Italy, the implementation of vaccination is the reason for the decline of measles incidence from 150/100,000 cases in the sixties to 5/100,000 in 2001 and to 0.3/100,000 in 2020. The relationship between measles vaccination coverage and hospitalization rate is inversely related [18]. However, vaccine implementation and the prevalence of positive antibodies are still insufficient to eradicate measles [19].

The effectiveness of vaccination against measles was evaluated in a large population of students attending courses of the School of Medicine of Padua University and subjected to health surveillance according to the law. The aim of the study was to ascertain the prevalence of positive antibody and the antibody titer according to vaccination schedule (one or two doses), vaccine type (vaccine containing measles attenuated virus alone or combined as MMR), and sex also. The first strength is the number of the enrolled students, who were all born in Italy and therefore subjected to the same vaccination schedule; the second strength is that all recruited students had a vaccination certificate released by the Public Health Office. Given these solid bases (population size and certainty about vaccination), the results obtained can establish a definitive point on the seroprevalence of antibodies in response to vaccination against measles. Reasons for weakness could be seen in the clear numerical difference between males and females (difference due to the predominant female presence in the degree courses of the health professions) and that the age of first administration is not the same for everyone but depends on the years in which were born. A further limitation could be the absence of some information on attitudes practices toward measles vaccination among the students, but this is a retrospective study based on data collected during health surveillance.

Three are the main results in this study: (1) to our knowledge, for the first time it is proved that measles vaccine alone appears somewhat more effective than MMR combination, and that one dose apparently induces a higher percentage of seropositivity and a higher antibody titer than two doses (unless one or both doses are measles alone); (2) there is high vaccine coverage in the younger generations; (3) vaccine, either one or two doses, induces a significantly higher prevalence of positive antibodies and higher antibody titer in females than in males.

The first evidence is not easily understood. A possible explanation has been recently suggested on a “negative” influence of the Rubini strain in MMR combination used for mumps between 1990 and 1995 [14], but at the moment, it remains unresolved. Nevertheless, the recent WHO position paper on measles vaccine [20] claims that “although vaccine-induced antibody concentrations decline over time and may become undetectable, immunological memory persists and, following exposure to measles virus, most people who have been vaccinated produce a protective immune response”. Although serum IgG measles titer declines during time, this does not happen for measles-specific neutralizing antibody titer [21], and two doses of measles vaccine results in high seropositivity, vaccine effectiveness, and T-cell response [22].

The second evidence highlights the significant effort made by health authorities to involve families in the vaccination program for the purpose to eradicate transmittable diseases such as measles, mumps, and rubella by achieving herd immunity. Regarding measles, the objective is complicated by the fact that herd immunity can only be achieved with a vaccination coverage over 95%, practically vaccinating all newborns [23].

The third evidence is that females have a significant higher percentage of positive response and higher antibody titer after vaccination than in males. Sex could be a variable in response to measles vaccination and disagree with other research [24] that do not support the fact that sex influences measles humoral and cellular immunity.

Sex differences in response to vaccine were recognized previously [25,26,27]; females develop a higher titer of neutralizing antibodies [28]. Inasmuch, if innate immunity is similar in both sexes, adaptive immunity is more pronounced in females [25]. Furthermore, differences in Toll-like receptor pathway and type I interferon induction may explain these differences [29], being several immune-related genes located on the X chromosome and play a pivotal role in immune competence [30]. Therefore, it is plausible that there are different immune responses if the vaccination is administered at different ages; measles vaccination is administered during the second year of life when adaptive immunity should be just developed. On the other hand, these differences are significant only if MMR (once or twice, indifferently) is adopted but not if measles vaccine alone was used.

The significant reduction in the circulation of the wild virus, as demonstrated by the low prevalence of positive antibodies in the unvaccinated younger student population, certainly reduces the possibility of natural boosters in vaccinated subjects resulting in a waning of circulating antibodies (but of immune memory also?) and a possible resurgence of measles outbreaks [31]. In addition, a booster dose is suggested in unprotected subjects after two doses of the vaccine, and a high seroconversion was observed [32].

The measles epidemic of 2017 in Italy prompted the government to enact a law to make 10 vaccines, including measles, mandatory from 0 to 16 years [5]. The mandatory effect has resulted in a significant increase in vaccination coverage [33,34], but this is not always understood and shared. An experiment of some interest began in the Veneto Region (Italy) in 2007 with the suspension of the vaccination obligation against poliomyelitis, diphtheria, tetanus, and hepatitis B, simultaneously investing in a health education campaign [35]. In any case, the study population, even if for the most part born and resident in Veneto, is not among the cohorts involved in the project.

Finally, the phenomenon of vaccine hesitancy is the cause of low vaccination coverage even among HCWs [36,37]. As previously reported [14], this phenomenon will certainly be much less evident in the near future given that our series shows a high vaccination coverage in what will be future HCWs.

## 5. Conclusions

Two are the main conclusions of this study: (i) females are more responsive to measles vaccination than males but only when MMR combined vaccine is used, and (ii) there is a better positive response and higher antibody titer if measles is administered alone (one or two dose), or measles alone plus MMR. Both evidences are intriguing, but the second is not easily to explain and remains an objective for further studies.

Despite the unavailability of the measles vaccine alone, it could be apposite that the pharmaceutical companies prepare and make available single doses to be used, in case as a booster, in HCWs, especially if the positivity against mumps and rubella is confirmed by the laboratory. Therefore, in this case, it would be interesting to study, in a time of low virus circulation, the differences in the antibody response between the measles vaccine alone and that in the MMR formulation.

The medical students (medicine, dentistry, and healthcare professions) will be future HCWs; then, particular attention must be given to the fact that they may have very close contact with patients affected by infectious diseases during their training and profession, mainly when transmitted by air. 

## Figures and Tables

**Figure 1 vaccines-09-00377-f001:**
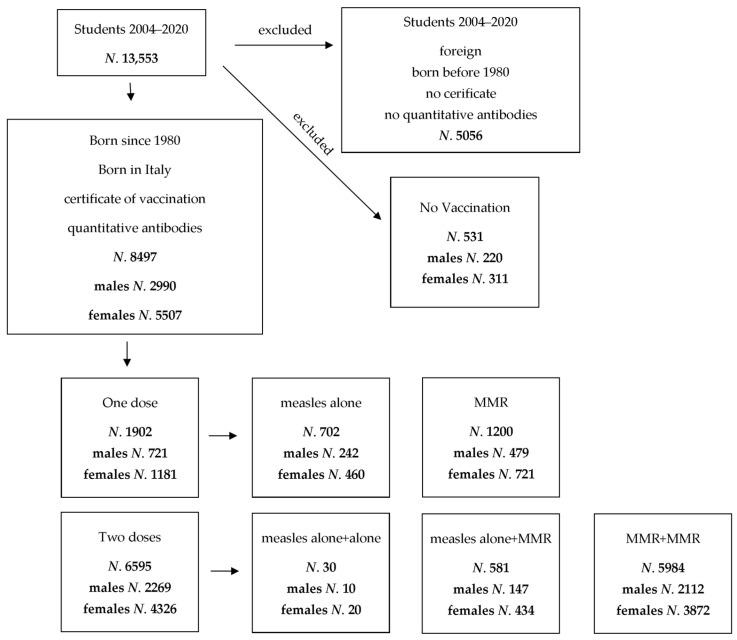
Criteria adopted to enroll students in the study. Only vaccinated students, born in Italy, who presented a vaccine certificate released by the Public Health Office and with quantitative measurement of measles antibodies were enrolled.

**Figure 2 vaccines-09-00377-f002:**
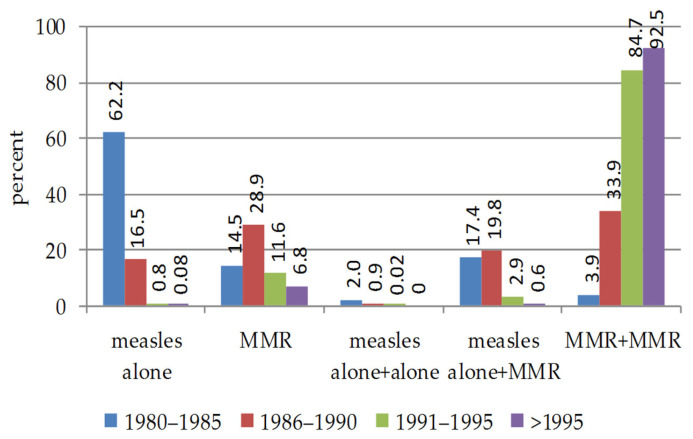
Percentage of vaccinated students according to adopted vaccination schedule. They are subdivided according to year of birth group. The Public Office certified that the measles vaccine was administered according to one dose schedule, alone or MMR, or two doses (two doses of measles alone, a dose of measles alone plus MMR, or two doses of MMR).

**Table 1 vaccines-09-00377-t001:** Characteristics of enrolled students according to graduate course and geographical origin.

Students	*N*.	Age at Analysis	Medicine and Surgery	Dentistry	Healthcare Professions	Northern Italy	Central Italy	Southern Italy	Veneto	Padua
		Mean ± SD	*N.*	*N.*	*N.*	*N.*	*N.*	*N.*	*N.*	*N.*
all	8497	21.1 ± 2.0	4385	252	3860	7939	162	396	7190	2777
males	2990	21.3 ± 2.0	1929	133	928	2747	68	175	2531	969
females	5507	21.0 ± 1.9	2456	119	2932	5192	94	221	4659	1808

**Table 2 vaccines-09-00377-t002:** Percentage of positive students and antibody titer according to vaccination schedule.

Vaccination Schedule	*N*.	Positives	%	Significance	Titer IU/mL	Significance
					Mean ± SD	
all one dose	1902	1663	87.4	^a^	1730.2 ± 1873.3	^aa^
measles alone	702	678	96.6	^b,c,d^	1824.2 ± 1673.1	^dd^
MMR	1200	985	82.1	^b,e,f^	1675.1 ± 1979.7	^ee,ff^
all two doses	6595	5144	78.0	^a^	1335.5 ± 1686.3	^aa^
measles alone + measles alone	30	29	96.7		1875.0 ± 1342.4	
measles alone + MMR	581	535	92.1	^c,e,g^	1668.0 ± 1697.9	^ee,gg^
MMR + MMR	5984	4580	76.5	^d^	1300.5 ± 1682.9	^dd^

Legend: meaning of superscript letters: ^a^ percentage of positives (*p* < 0.0001) and ^aa^ titer (*p* < 0.0001) after one dose vs. two doses; ^b^ percentage of positives (*p* < 0.0001) after one dose of measles alone vs. one dose of MMR; ^c^ percentage of positives (*p* = 0.0065) after one dose of measles alone vs. two doses of measles alone plus MMR; ^d^ percentage of positives (*p* < 0.0001) and ^dd^ titer (*p* < 0.0001) after one dose of measles alone vs. two doses of MMR; ^e^ percentage of positives (*p* < 0.0001) and ^ee^ titer (*p* < 0.0001) after two doses of measles alone plus MMR vs. one dose of MMR; ^f^ percentage of positives (*p* = 0.0003) and ^ff^ titer (*p* < 0.0001) after one dose of MMR vs. two doses of MMR; ^g^ percentage of positives (*p* < 0.0001) and ^gg^ titre (*p* < 0.0001) after two doses of measles alone plus MMR vs. two doses of MMR.

**Table 3 vaccines-09-00377-t003:** Percentage of positive students and antibody titer according to sex and vaccination schedule.

Doses	Vaccination Schedule	Sex	*N*.	Positives	%	Significance	Titer IU/mL	Significance
							Mean ± SD	
1	all	males	721	602	83.5		1528.3 ± 1743.4	
1	all	females	1181	1061	89.8	^a^	1853.4 ± 1938.8	^aa^
1	measles alone	males	242	234	96.7		1789.8 ± 1806.1	
1	measles alone	females	460	444	96.5		1842.4 ± 1600.5	
1	MMR	males	479	368	76.8		1396.1 ± 1697.5	
1	MMR	females	721	617	85.6	^b^	1860.5 ± 2127.7	^bb^
2	all	males	2269	1689	74.4		1247.5 ± 1629.6	
2	all	females	4326	3455	79.9	^c^	1381.6 ± 1713.6	^cc^
2	measles alone+alone	males	10	10	100.0		1906.0 ± 1349.2	
2	measles alone+alone	females	20	19	95.0		1859.5 ± 1373.9	
2	measles alone+MMR	males	147	136	92.5		1668.4 ± 1461.1	
2	measles alone+MMR	females	434	399	91.9		1667.9 ± 1772.5	
2	MMR+MMR	males	2112	1543	73.1		1215.1 ± 1637.8	
2	MMR+MMR	females	3872	3037	78.4	^d^	1347.0 ± 1705.5	

Legend: meaning of superscript letters: ^a^ percentage of positives (*p* = 0.0004) and ^aa^ titer (*p* = 0.0010) after one dose, females vs. males; ^b^ percentage of positives (*p* = 0.0067) and ^bb^ titer (*p* = 0.0137) after one dose of MMR, females vs. males; ^c^ percentage of positives (*p* < 0.0001) and ^cc^ titer (*p* = 0.0110) after two doses, females vs. males; ^d^ percentage of positives (*p* = 0.0001) after two doses of MMR, females vs. males.

**Table 4 vaccines-09-00377-t004:** Multiple linear regression among the logarithmic transformation of antibody titer of measles in vaccinated students and the independent variables sex, age of vaccination (or first dose of vaccine), time between vaccination, and antibody measurement and typology of vaccination (see legend). Statistically significant results are in bold.

	b	SE	t	*p*
Intercept	3.194	0.044	71.887	<0.0001
sex	0.060	0.101	5.985	<0.0001
typology 1	0.000			
typology 2	−0.189	0.022	−8.563	<0.0001
typology 3	−0.008	0.084	−0.100	0.92
typology 4	−0.147	0.031	−4.678	<0.0001
typology 5	−0.312	0.026	−11.714	<0.0001
age 1st dose	0.000	0.000	6.774	<0.0001
time	−0.000	0.000	−4.584	<0.0001

Legend: typology 1 signifies one dose with measles alone (reference typology), typology 2 signifies one dose of MMR, typology 3 signifies two doses of measles alone, typology 4 signifies one dose of measles alone plus one dose of MMR, typology 5 signifies two doses of MMR. In the header: b = slope, SE = standard error of b, t = *t*-test; that is, the relationship between b and SE.

**Table 5 vaccines-09-00377-t005:** Percentage of positive antibodies and antibody titer according to doses of vaccine and sex. The *p* values refer to the statistical comparison between males and females.

**Year of Birth**		***N.***	**Positives**	**%**	***p***	**Titer IU/mL**	***p***
**One Dose**						**Mean ± SD**	
1980–1985	all	454	448	98.7		1997.6 ± 1818.7	
	males	157	153	97.5		1779.4 ± 1630.6	
	females	297	295	99.3	0.2182	2113.0 ± 1903.2	0.0514
1986–1990	all	832	798	95.9		1822.1 ± 1756.2	
	males	293	275	93.9		1798.0 ± 1923.9	
	females	539	523	97.0	0.0428	1835.3 ± 1659.7	0.7795
1991–1995	all	441	286	64.9		1477.1 ± 2167.6	
	males	189	110	58.2		1097.9 ± 1603.9	
	females	252	176	69.8	0.0150	1761.5 ± 2473.9	0.0007
after 1995	all	175	131	74.9		1236.6 ± 1576.0	
	males	82	64	78.0		1075.6 ± 1248.0	
	females	93	67	72.0	0.4597	1378.5 ± 1811.9	0.1955
**Year of Birth**		***N.***	**Positives**	**%**	***p***	**titer IU/mL**	***p***
**Two Doses**						**Mean ± SD**	
1980–1985	all	138	135	97.8		1733.2 ± 1465.6	
	males	43	40	93.0		1630.2 ± 1147.9	
	females	95	95	100.0	0.0485	1779.8 ± 1592.1	0.5334
1986–1990	all	1000	960	96.0		1717.2 ± 1713.4	
	males	317	299	94.3		1564.4 ± 1477.7	
	females	683	661	96.8	0.0946	1788.1 ± 1809.1	0.0388
1991–1995	all	3103	2191	70.6		1377.4 ± 1855.5	
	males	1114	743	66.7		1325.8 ± 1842.2	
	females	1989	1448	72.8	0.0004	1406.3 ± 1862.7	0.2454
after 1995	all	2354	1858	78.9		1094.7 ± 1388.2	
	males	795	607	76.4		990.7 ± 1329.4	
	females	1559	1251	80.2	0.0327	1147.7 ± 1414.7	0.0081

## Data Availability

Raw data are available on request from the corresponding author.

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
