# Peer review of "Vaccination and Immunity toward Measles: A Serosurvey in Future Healthcare Workers"

_vaccines, 2021, doi:10.3390/vaccines9040377_

Round 1

Reviewer 1 Report

Estimated Authors,

firstly, thank you for the opportunity to review your paper on a serosurvey among future healthcare workers from the university of Padua. The present paper is of clear interest and significance, but - in my opinion, several improvements are required before the eventual publication. More precisely:

1) please include a detailed table including demographic data of study participants (age, sex, number of doses received, etc); some of the data that may improve the understanding of your sample are (in fact) "not showed", and a preliminary table may be a possible option.

2) Table 1 --> 5 are quite overcrowded and/or confusing and should be redesigned in order to improve the overall readability of their content. In particular, by dichotomizing Table 4 and Table 5 in two distintive panels with a very different content may confuse your reader, particularly when dealing with the comparisons. In this regard...

3) ... please be aware of the multiple independent comparisons you performed, that without a post-hoc correction may have inflated the potential p values you reported. Have you considered, for example, a Bonferroni's correction?

4) In table 3, the equation reported above the single panels (e.g. antibody titre = 890.395835 +134.501966 sex +0.211522 age 1st dose +0.078826 age 2nd dose ‐0.024407 time since the 2nd dose) may be somewhat misleading the for majority of readers and it not really informative; it may be removed.

5) In the discussion section, you "grazed" a significant issue, i.e. the potentially reduced seroprevalence of IgG among subjects who where vaccinated by means of combined formulates compared to monovalent vaccines. This is quite interesting, albeit disappointing in a Public Health perspective: therefore, you should discuss in further details how this issue, if confirmed, may impair vaccination campaigns and vaccinations schedules in the workplaces.

6) in the discussion section, you have not reported on the "Veneto exception" regarding vaccination schedules; in facts, until recently, Veneto Region (where Padua resides) was a sort of "laboratory" for improving vaccination rates without mandate. How many of the students you included in your analyses were from Veneto region? It would be particularly interesting, as the falling vaccination rates for measles were among the reasons that promoted current vaccination policies.

7) have you performed any assessment of Knowledge Attitudes Practices towards measles vaccination among the students you assessed?

Thank you for the considerable efforts to collect and summarize this extensive amount of data: I'm confident that the aforementioned requirement (and those from other reviewer as well) may be quite simply addressed and will eventually improve the overall quality of your very interesting study.

Author Response

Comments and Suggestions for Authors

Estimated Authors,

firstly, thank you for the opportunity to review your paper on a serosurvey among future healthcare workers from the university of Padua. The present paper is of clear interest and significance, but - in my opinion, several improvements are required before the eventual publication. More precisely:

We are grateful to the reviewer for the suggestion. Point by point:

1) please include a detailed table including demographic data of study participants (age, sex, number of doses received, etc); some of the data that may improve the understanding of your sample are (in fact) "not showed", and a preliminary table may be a possible option.

Reply: We modified Figure 2 (without gender breakdowns) and added Table 2 with the characteristics of the study population, including their geographic origin. The Table 1 of the original text has been removed.

2) Table 1 --> 5 are quite overcrowded and/or confusing and should be redesigned in order to improve the overall readability of their content. In particular, by dichotomizing Table 4 and Table 5 in two distinctive panels with a very different content may confuse your reader, particularly when dealing with the comparisons. In this regard...

Reply: We are grateful to the reviewer for the suggestion. Table 4 and 5 are now Table 2 and 3 and statistics are indicated in the legend then panel A and B have been removed.

3) ... please be aware of the multiple independent comparisons you performed, that without a post-hoc correction may have inflated the potential p values you reported. Have you considered, for example, a Bonferroni's correction?

Reply: Bonferroni’s correction was applied.

4) In table 3, the equation reported above the single panels (e.g. antibody titre = 890.395835 +134.501966 sex +0.211522 age 1st dose +0.078826 age 2nd dose ‐0.024407 time since the 2nd dose) may be somewhat misleading the for majority of readers and it not really informative; it may be removed.

Reply: All the tables have been removed and replaced by a single table (now Table 4).

5) In the discussion section, you "grazed" a significant issue, i.e. the potentially reduced seroprevalence of IgG among subjects who were vaccinated by means of combined formulates compared to monovalent vaccines. This is quite interesting, albeit disappointing in a Public Health perspective: therefore, you should discuss in further details how this issue, if confirmed, may impair vaccination campaigns and vaccinations schedules in the workplaces.

Reply: It is clear that pharmaceutical companies have no economic interest in producing single vaccines, so it remains a mere academic conjecture, as indicated in the second paragraph of the conclusions.

6) in the discussion section, you have not reported on the "Veneto exception" regarding vaccination schedules; in facts, until recently, Veneto Region (where Padua resides) was a sort of "laboratory" for improving vaccination rates without mandate. How many of the students you included in your analyses were from Veneto region? It would be particularly interesting, as the falling vaccination rates for measles were among the reasons that promoted current vaccination policies.

Reply: “Veneto exception” started in 2007, then no students had been vaccinated under this exception. On the other hand exception regarded mandatory vaccination such as against poliomyelitis, tetanus, diphtheria and hepatitis B. Exception derived also by the large compliance versus non mandatory vaccination. In any case, the cases presented to us are among the cohorts involved in the project. Considerations are added in the text in this regard.

7) have you performed any assessment of Knowledge Attitudes Practices towards measles vaccination among the students you assessed?

Reply: Being a retrospective study based on data collected during health surveillance, no questionnaires were proposed to assess knowledge attitudes practices towards measles vaccination among the students. The same considerations were added in the discussion section at the end of the second paragraph.

Thank you for the considerable efforts to collect and summarize this extensive amount of data: I'm confident that the aforementioned requirement (and those from other reviewer as well) may be quite simply addressed and will eventually improve the overall quality of your very interesting study.

Reviewer 2 Report

I was invited to revise the paper entitled "Vaccination and Immunity towards Measles: a Serosurvey in Future Healthcare Workers". It was a retrospective study among Italian students of Health care University Courses aimed to evaluate immunity agaist Measles. The topic is interesting and this paper studied a large cohort of students from different settings. However I have same major observation to highlight:

  • Authors should report a table presenting baseline characteristics of all enrolled patients;
  • Table 1 is unreadable and its significance is unclear. Did Authors wanted to highlight gender differences? was It the main aim? I don't think so;
  • Authors should present table 4 before tables 2 and 3. Titres should be presented before regression analyses;
  • Authors did not tested continous variables for normal distribution. Titres are clearly not normally distributed so linear regression analysis cannot be performed;
  • Authors should sum panel a and b of table 4;
  • Authors should correct all analyses for multiple comparison;
  • Discussion was poor. In my knowledge for Italian law vaccination for HCWs are not mandatory. Authors should discuss about HCWs vaccine hesitancy towards MMR vaccination, as presented by recent papers ( 10.3390/vaccines8020248 and 10.15167/2421-4248/jpmh2019.60.1.1097);
  • It is unclear how Authors selected dependent variables in multivariable models;
  • Authors should adjust multivariable models by university degree attended and by region of origin: different place of origin probably had different vaccination schedule that impacted on the age of vaccine uptake;

Minor comments:

  • Authors should replace the word "alone" in figures and tables with "Measles alone";
  • Authors have to show exact p-value instead of n.s..

Author Response

Reviewer 2

Comments and Suggestions for Authors

I was invited to revise the paper entitled "Vaccination and Immunity towards Measles: a Serosurvey in Future Healthcare Workers". It was a retrospective study among Italian students of Health care University Courses aimed to evaluate immunity agaist Measles. The topic is interesting and this paper studied a large cohort of students from different settings. However I have same major observation to highlight:

We are grateful to the reviewer for the suggestion. Point by point:

Authors should report a table presenting baseline characteristics of all enrolled patients;

Reply: As also replied to Reviewer 1, a Table (Table 1) was inserted with the data of the population involved in the study.

Table 1 is unreadable and its significance is unclear. Did Authors wanted to highlight gender differences? was It the main aim? I don't think so;

Reply: We agree with the reviewer's observation. Table 1 has been deleted.

Authors should present table 4 before tables 2 and 3. Titres should be presented before regression analyses;

Reply: Tables were swapped as required. Tables 4 and 5 of the original manuscript became Tables 2 and 3 while Tables 2 and 3 (original) were summarized in Table 4.

Authors did not tested continous variables for normal distribution. Titres are clearly not normally distributed so linear regression analysis cannot be performed;

Reply: Regression analyses are removed, being in this case unimportant. Antibody titre is not normally distributed then logarithmic conversion has been used also in multiple analysis.

Authors should sum panel a and b of table 4;

Reply: We are grateful to the reviewer for the suggestion. Table 4 and 5 are now Table 2 and 3 and statistics are indicated in the legend then panel A and B have been removed.

Authors should correct all analyses for multiple comparison;

Reply: Antibody titre is not normally distributed then logarithmic conversion has been used also in multiple analysis.

Discussion was poor. In my knowledge for Italian law vaccination for HCWs are not mandatory. Authors should discuss about HCWs vaccine hesitancy towards MMR vaccination, as presented by recent papers ( 10.3390/vaccines8020248 and 10.15167/2421-4248/jpmh2019.60.1.1097);

Reply: We implemented discussion as suggested.

It is unclear how Authors selected dependent variables in multivariable models;

Reply: We hope that new version allows to clarify what is suggested.

Authors should adjust multivariable models by university degree attended and by region of origin: different place of origin probably had different vaccination schedule that impacted on the age of vaccine uptake;

Reply: The vaccination schedule should be common throughout the national territory. However, this is not written in the stone and therefore varies over time according to new scientific findings. Figure 2 highlights this variability over time and the distribution between groups by year of birth.

Minor comments:

Authors should replace the word "alone" in figures and tables with "Measles alone";

Reply: in Tables measles alone has been inserted

Authors have to show exact p-value instead of n.s..

Reply: p-values have been inserted

Reviewer 3 Report

The study is really interesting and complete, I think we can simplify the text a little to make it more understandable to readers, my comments are directly included in the PDF

Author Response

Reviewer 3

Comments and Suggestions for Authors

The study is really interesting and complete, I think we can simplify the text a little to make it more understandable to readers, my comments are directly included in the PDF

We are grateful to the reviewer for the suggestion. Point by point:

abstract: We need to add a more information on how the data is obtained. We also need the main results in abstract

Reply: What was requested has been implemented.

introduction: introduction could be a more developed, especially this part between vaccination and healthcare worker.

Reply: This part has been implemented.

materials and methods, population: it is necessary to explain more this flow chart of recruitment

Reply: We hope that the explanation also in the header of Figure 1 is sufficiently explanatory.

results: I find that we get a little lost in all these tables which makes us lose the key message of the authors

Reply: The multiple regression tables have been removed and replaced by a single all-encompassing table (Table 4).

Table 1: explanations of this table 1 should be further developed in the text

Reply: According to reviewer 1, Table 1 was deleted.

discussion: it lacks a clearly defined and developed limit of the study paragraph

Reply: At the end of the second paragraph of the discussion, we added a brief consideration on the possible weakness of the research.

Round 2

Reviewer 1 Report

Estimated Authors,

I've appreciated the considerable efforts you paid in order to cope with my previous recommendations.

Only some minor remarks (excuse me) about Table 2 and 4:

Table 2: it is properly informative, and improves the understanding of your data; however, some improvements in its formatting are requested, particularly in order to better discriminate data on healthcare profession (columns MS, DE, HP) from those on the geographical origins of the participants (columns NI to SI); in this regard, column "VENETO" could be moved either at the far left of at the far right of this subsection in order to highlight its significance. 

Table 4, its formatting is somewhat awkward, but I'm confident you can fix it quite rapidly.

Author Response

Estimated Authors,

I've appreciated the considerable efforts you paid in order to cope with my previous recommendations.

Only some minor remarks (excuse me) about Table 2 and 4:

We are grateful to the reviewer for further suggestions. Point by point:

Table 2: it is properly informative, and improves the understanding of your data; however, some improvements in its formatting are requested, particularly in order to better discriminate data on healthcare profession (columns MS, DE, HP) from those on the geographical origins of the participants (columns NI to SI); in this regard, column "VENETO" could be moved either at the far left of at the far right of this subsection in order to highlight its significance.

Reply: We think the reviewer is referring to Table 1. Veneto and PD (Padua) have been moved to the far right. The legend, as requested by reviewer 2, has been removed and the abbreviations have been written in full in the header of the Table.

Table 4, its formatting is somewhat awkward, but I'm confident you can fix it quite rapidly.

Reply: We are sorry for what happened to the table formatting, which has been corrected.

Reviewer 2 Report

I was invited to review the revised version of the paper entitled "Vaccination and Immunity towards Measles: a Serosurvey in Future Healthcare Workers". Authors improved the paper accordingly to reviewers comments. There are some minor observation to be addressed:

  • Footer of table 1 should be removed. Are useless;
  • In statistical analysis section describe extensively p-value correction procedure;
  • Titres in table 3 should be presented after logaritmic transformation;
  • In table 4 report only three decimal number;
  • In table 5 remove the sign =;

Author Response

Comments and Suggestions for Authors

I was invited to review the revised version of the paper entitled "Vaccination and Immunity towards Measles: a Serosurvey in Future Healthcare Workers". Authors improved the paper accordingly to reviewers comments. There are some minor observation to be addressed:

Footer of table 1 should be removed. Are useless;

Reply: The legend has been removed.

In statistical analysis section describe extensively p-value correction procedure;

Reply: In the statistics section we have better detailed the p value correction procedures.

Titres in table 3 should be presented after logaritmic transformation;

Reply: We think the reviewer is referring to Table 4. The required information has been entered in the header of the table.

In table 4 report only three decimal number;

Reply: As requested, only three decimal places have been entered.

In table 5 remove the sign =;

Reply: sign = has been removed.
